# Statistical Characteristics and Composite Environmental Conditions of Explosive Cyclones over the Japan Sea and Kuroshio/Kuroshio Extension

**Shuqin Zhang** [1,2,3,4], **Gang Fu** [5], **Yu Zhang** [1,2,3,4], **Jianjun Xu** [1,2], **Yufeng Xue** [1,2,3,4], **Ruoying Tang** [6], **Xiaoyu Gao** [7], **Chunlei Liu** [1,2,*] and **Jingchao Long** [1,2,3,4,*]

1   CMA-GDOU Joint Laboratory for Marine Meteorology, Zhanjiang 524088, China; zhangsq@gdou.edu.cn (S.Z.); zhangyu2@gdou.edu.cn (Y.Z.); jxu@gdou.edu.cn (J.X.); xueyf@gdou.edu.cn (Y.X.)
2   South China Sea Institute of Marine Meteorology, Guangdong Ocean University, Zhanjiang 524088, China
3   College of Ocean and Meteorology, Guangdong Ocean University, Zhanjiang 524088, China
4   Key Laboratory of Climate, Resources and Environment in Continental Shelf Sea and Deep Sea of Department of Education of Guangdong Province, Guangdong Ocean University, Zhanjiang 524088, China
5   College of Oceanic and Atmospheric Sciences, Ocean University of China, Qingdao 266100, China; fugang@ouc.edu.cn
6   College of Coastal Agricultural Sciences, Guangdong Ocean University, Zhanjiang 524088, China; rtang@gdou.edu.cn
7   State Key Laboratory of Severe Weather, Chinese Academy of Meteorological Sciences, Beijing 100081, China; gaoxy@cma.gov.cn
*   Correspondence: liuclei@gdou.edu.cn (C.L.); longjc@gdou.edu.cn (J.L.)

**Abstract:** Statistical characteristics and composite synoptic-scale environmental conditions of explosive cyclones (ECs) over the Japan Sea and Kuroshio/Kuroshio Extension are examined and compared using ERA5 atmospheric reanalysis to give a better understanding of their differences. ECs over the Japan Sea frequently occur in late autumn and early winter and those over the Kuroshio/Kuroshio Extension mainly occur in winter and early spring. The maximum deepening rate, minimum central sea level pressure and explosive-developing lifetime of ECs over the Kuroshio/Kuroshio Extension are generally larger, lower and longer, respectively, than those over the Japan Sea. ECs over the Kuroshio/Kuroshio Extension formed over the East China Sea tend to develop more rapidly, and weak and moderate ECs generally begin to develop explosively over the sea to the east of the Japan Islands, while the strong and super ECs over the sea to the south of Japan Islands have longer explosive-developing tracks. Composite analysis shows that synoptic-scale environmental conditions favoring rapid EC development over these two regions are significantly different. ECs over the Japan Sea have stronger baroclinicity and cyclonic vorticity, but weaker water vapor convergence and upper-level jet stream than those over the Kuroshio/Kuroshio Extension. The key factor contributing to the baroclinicity is the cold air intrusion over the Japan Sea and the strong warm current heating over the Kuroshio/Kuroshio Extension. The potential vorticity shows anomalies in upper and low levels for both EC areas and extends further downwards over the Japan Sea.

**Keywords:** explosive cyclone; statistical composite analysis; environmental condition; Japan Sea; Kuroshio/Kuroshio Extension

## 1. Introduction

Explosive cyclones (ECs) are rapidly developing cyclonic systems whose central sea level pressure (SLP) drops more than or equal to 24 hPa within 24 h when adjusted geostrophically to 60° N [1]. Zhang et al. [2] adjusted the latitude to 45° N and set the threshold to 12 hPa/12 h, so one "Bergeron", defined as the change of a 24 hPa/24 h at 60° N in Sanders and Gyakum [1], was modified into a 12 hPa/12 h at 45° N (hereinafter referred to as the modified Bergeron: MBer). In addition, Zhang et al. [2] classified these ECs into four categories using a K-means clustering algorithm based on the maximum deepening

rate (MDR): super ($\geq$2.30 MBer), strong (1.70–2.29 MBer), moderate (1.30–1.69 MBer) and weak (1.00–1.29 MBer). ECs usually lead to extremely bad weather such as severe winds and heavy precipitation that can potentially cause serious losses of life and property [3–5], so they are thought to be one of the most dangerous weather systems and are known as the "Meteorological bomb" [1].

Climatological features of ECs have been investigated by many previous studies [1,6–11] and the results show that ECs occur mostly over the mid-latitude ocean in cold season. The northwestern Pacific is one of the regions with frequent EC occurrence [2,12]. Chen et al. [13] and Yoshida and Asuma [8] pointed out that the Japan Sea and the northwestern Pacific along the Kuroshio/Kuroshio Extension are two favorable areas for rapid EC development. Chen et al. [13] found that ECs occurred most frequently in December, January and March. Yoshida and Asuma [8] and Zhang et al. [2] suggested that the occurrence frequency of ECs shows evident seasonal variations for various regions over the northern Pacific; ECs over both the Japan Sea and the Okhotsk Sea peaked in November, whereas the maximum EC frequency over the northeastern Pacific is in January in Yoshida and Asuma [8] and December in Zhang et al. [2]. However, other statistical features over the Japan Sea and Kuroshio/Kuroshio Extension, such as the minimum SLP, explosive-developing lifetime, etc., are rarely examined.

Large-scale atmospheric environmental conditions have significant effects on EC development. Sanders and Gyakum [1] suggested that ECs frequently occur within or at the poleward side of the upper-level jet stream [2,14] and the downstream of a 500 hPa trough [13,15,16]. Yoshida and Asuma [8] showed that the cold air mass originating from the Asian Continent resulted in strong baroclinicity over the northwestern Pacific, favoring rapid EC development [9]. The warm current over the Kuroshio/Kuroshio Extension in the northwestern Pacific provides a favorable environment for frequent EC occurrences, due to its heat and moisture supply, facilitating unstable conditions within the atmospheric boundary layer and intensifying baroclinicity in the lower troposphere [17–19]. Several factors including the low-level baroclinicity [20–22], latent heat release [23–27], vorticity advection [28,29], upper-level jet stream [21,30] and PV (potential vorticity, see Section 2.2.2) [5,22,31–33] contribute to the rapid EC development. The relative importance of these factors varies strongly from region to region, due to different larger-scale environmental conditions [2,14,20,21,34].

The Japan Sea and the Kuroshio/Kuroshio Extension are regions of high frequency of EC occurrences [2]. Although some previous studies have focused on the EC development over these two areas [8,13,18,35], the detailed statistical features and environmental conditions, such as the characteristics of the minimum central SLP, explosive-developing lifetime, water vapor, PV, etc., have seldom been analyzed. A detailed study of these features should be conducted using high resolution data for further understanding of EC development over the Japan Sea and Kuroshio/Kuroshio Extension. The purpose of this study is to investigate and compare the statistical characteristics and synoptic-scale environmental conditions of ECs over the Japan Sea and Kuroshio/Kuroshio Extension. ECs during 15-year (2000–2015) cold-seasons (October–April) are identified and tracked using an objective method and the high resolution data of ERA5 (the fifth generation ECMWF ReAnalysis [36]). The paper is organized as follows. Section 2 describes the data and methods. The statistical characteristics and the synoptic-scale environmental conditions of ECs over the Japan Sea and Kuroshio/Kuroshio Extension are investigated and compared in Section 3. Finally, Section 4 presents discussion and conclusions.

## 2. Data and Methods

### 2.1. Data

Hourly variables, including the geopotential height, horizontal wind, temperature, relative humidity and SLP, are from ERA5, provided by the European Centre for Medium-Range Weather Forecasts (ECMWF). ERA5 data are produced using 4D-Var data assimilation in CY41R2 of ECMWF's Integrated Forecast System (IFS) [36]. ERA5 data during 15-year (2000–2015) cold-seasons (October–April) are downloaded with 0.25° × 0.25° horizontal resolution and 37 vertical levels from 1000 to 1 hPa. The mean SLPs are used to identify ECs and the geopotential height, horizontal wind, temperature, and relative humidity are employed to examine the composite synoptic-scale environmental conditions.

### 2.2. Methods

#### 2.2.1. Definition of EC

The definition of EC modified by Zhang et al. [2] is used in the present study, i.e., a cyclone with central SLP drop greater than or equal to 12 hPa within 12 h after adjusting geostrophically to 45° N. The deepening rate of the cyclone SLP (DR in MBer) is calculated using the following equation:

$$DR = \left( \frac{P_{t-6} - P_{t+6}}{12} \right) \times \left( \frac{\sin 45°}{\sin \frac{\phi_{t-6} + \phi_{t+6}}{2}} \right) \tag{1}$$

where $t$ is the time in hours, P the central SLP, and $\varphi$ the latitude of the cyclone center. Subscripts "$t - 6$" and "$t + 6$" represent times 6 h before and after the present time $t$, respectively.

#### 2.2.2. Relative Vorticity and PV Calculation

The relative vorticity can be calculated using the following Equation:

$$\zeta = \frac{\partial v}{\partial x} - \frac{\partial u}{\partial y} \tag{2}$$

where $\zeta$ is the relative vorticity, and $u$ and $v$ are the zonal and meridional winds, respectively.

The PV can be calculated by

$$PV = -g \left( \frac{\partial \theta}{\partial p} \right) (\zeta + f) \tag{3}$$

where g is gravitational acceleration, $\theta$ is the potential temperature, $p$ is the pressure, $\zeta$ is the relative vorticity and $f$ is the Coriolis parameter.

#### 2.2.3. Identification of ECs

ECs during 15-year (2000–2015) cold-seasons (October–April) are identified through an objective approach using ERA5 data. The cyclone was automatically detected and tracked by the minimum SLP, an algorithm developed by Hart [37]. The EC is then identified using the modified definition of Zhang et al. [2] described above. The main steps are as follows:

(1)　Cyclone detection

A cyclone is defined using the following criteria: (1) a local minimum SLP less than 1020 hPa within a 5° × 5° box, (2) a lifespan of at least 24 h, and (3) a minimum SLP gradient of 2 hPa within the 5° × 5° box. This box is then moved across the domain, partially overlapping the previous box location to ensure detection of cyclones that fall at the box edge.

(2)    Cyclone tracking

The minimum SLPs after two consecutive analyses (cyclone A at time $t - \Delta t$ and cyclone B at time $t$) separated by a distance $\Delta d$ are deemed to be the same cyclone at those two consecutive times only if the following tracking criteria are met: (1) $\Delta t = 3$ h, (2) cyclone B is the closest cyclone at time $t$ to where cyclone A was at time $t$-$\Delta t$, (3) $V$ (implied movement of the cyclone) = $\Delta d/\Delta t < 40$ m s$^{-1}$, and (4) $\Delta d < \Delta d_{MAX}$ (maximum allowed movement over $\Delta t$), where $\Delta d_{MAX} =$ Max (500 km, $3 \times \Delta t \times V_{t-\Delta t}$).

(3)    EC identification

Once the life history of each cyclone is obtained, the modified definition of Zhang et al. [2] is used to identified ECs by checking if the deepening rate is greater than or equal to 1 MBer.

### 2.2.4. Composite Analysis

In order to better understand and compare the synoptic-scale environmental conditions for EC development over the Japan Sea and Kuroshio/Kuroshio Extension, a composite analysis using the geographically fixed coordinate (20–60° N, 110–180° E) is conducted in this study. The maximum-deepening-rate moment of ECs is focused on in the composite analysis. Figure 1 shows the frequency of ECs with the maximum-deepening-rate position in a 3° × 3° grid box over 15-year (2000–2015) cold-seasons (October–April) in the Northern Pacific. The specified 3° × 3° box makes it easier to distinguish ECs over the Japan Sea, the Okhotsk Sea and the northwestern Pacific, and can also show characteristics of the geographic distribution clearly, compared with 4° × 4° or 5° × 5° boxes (not shown). The highest EC frequencies can be found over the Japan Sea and Kuroshio/Kuroshio Extension, with the maximum number of 17 and 20, respectively. ECs over these two regions surrounded by green lines are selected for analysis. There are 75 ECs over the Japan Sea and 129 ECs over the Kuroshio/Kuroshio Extension. The anomaly is also calculated relative to a 15-day running mean field. The 15-day mean field represents the time scale of the seasonal transition and longer phenomena, the anomaly indicating the related synoptic-scale disturbance of ECs. The ratio between anomaly and its standard error (defined as RAS) shows whether the anomaly is larger than the standard error for ECs over the Japan Sea and Kuroshio/Kuroshio Extension.

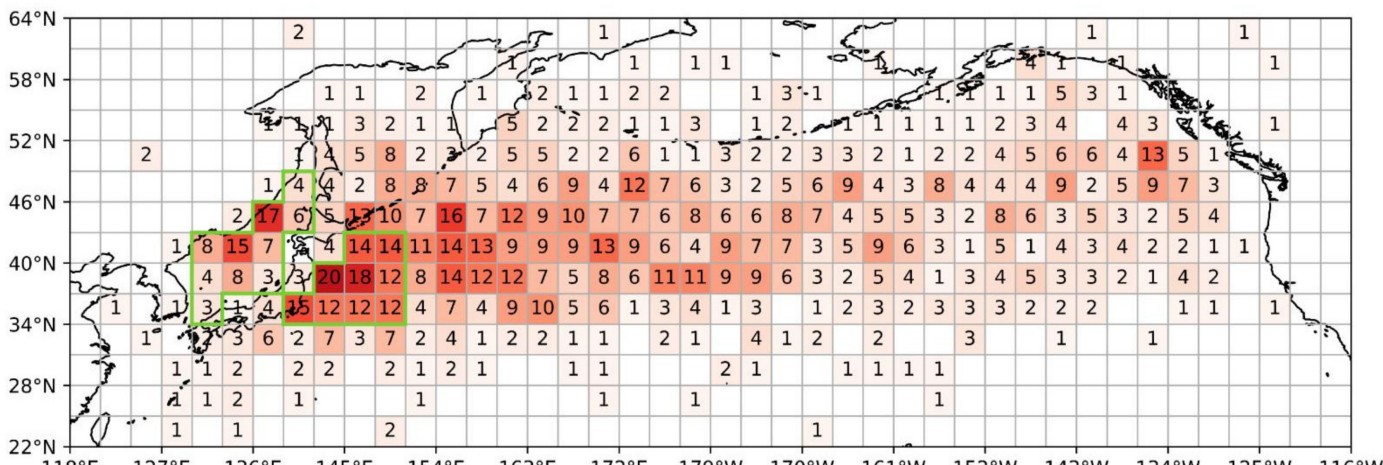

**Figure 1.** Frequency of ECs with the maximum-deepening-rate position within the 3° × 3° box over the northern Pacific during 15-year (2000–2015) cold-seasons (October–April). ECs in regions outlined by green lines over the Japan Sea and Kuroshio/Kuroshio Extension are selected for the composite analysis.

## 3. Results

### 3.1. Statistical Characteristics

3.1.1. Monthly Frequency

Figure 2 is the monthly frequency of ECs over the Japan Sea and Kuroshio/Kuroshio Extension. Over the Japan Sea, the highest frequency is 19 in December and there are only 6 ECs in October and 4 ECs in April. The frequency jumps from October to November and the monthly frequencies in January, February and March are about half of those in November and December. For ECs over the Kuroshio/Kuroshio Extension, the peak frequency is 34 in March and 27 in December, and October has the smallest number of 3. The frequency increases sharply from October to December and decreases slowly from December to February, but increases again in March and then a sudden drop in April follows. The patterns of frequency distribution over these two areas are significantly different, with 45.3% ECs over the Japan Sea occurring in late autumn (November) and early winter (December) and 82.2% ECs over the Kuroshio/Kuroshio Extension occurring in winter (December, January and February) and early spring (March). The monthly frequencies of ECs over these two areas show significant differences from Yoshida and Asuma [8]. They showed that the frequency peaked in November over the Japan Sea and Okhotsk Sea and in January over the Northwestern Pacific.

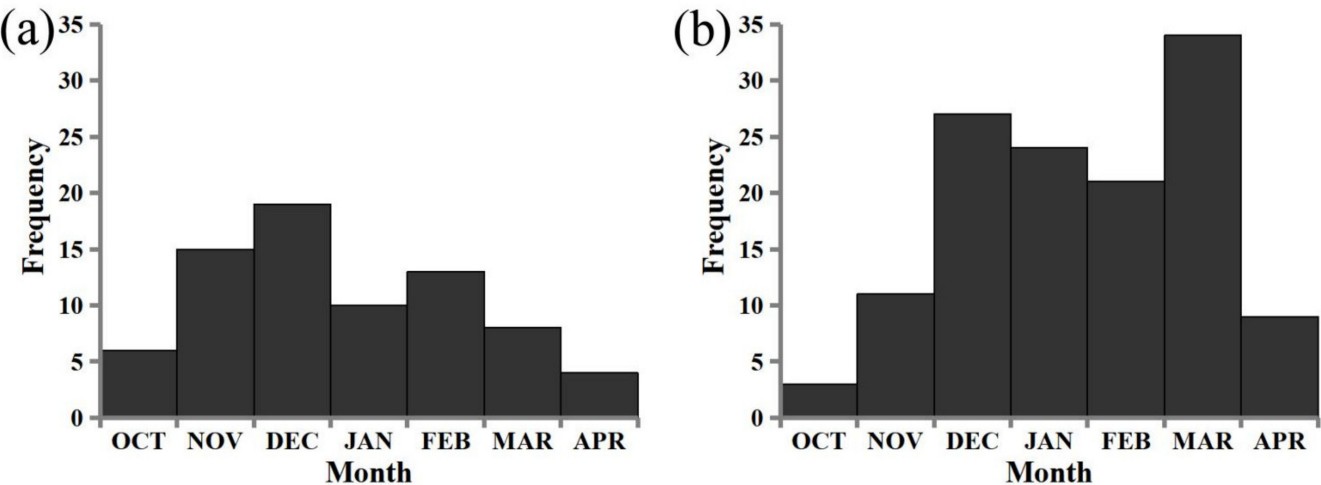

**Figure 2.** The monthly frequency of ECs over (**a**) the Japan Sea and (**b**) Kuroshio/Kuroshio Extension.

3.1.2. Frequency of MDR

The EC frequency at different MDR over the Japan Sea and Kuroshio/Kuroshio Extension is plotted in Figure 3. The frequency generally decreases along with the increase of the MDR, except that there is an increase from 1.1 MBer to 1.4 MBer for EC over the Kuroshio/Kuroshio Extension. The MDR for ECs over the Japan Sea is generally smaller than 1.7 MBer, with 41 (54.7%) weak and 26 (34.7%) moderate ECs. There are 7 (9.3%) strong and 1 (1.3%) super ECs. Most of the ECs over the Kuroshio/Kuroshio Extension are also less than 1.7 MBer, with 37 (28.7%) weak and 49 (40.0%) moderate ECs, and the percentage of moderate ECs is larger than that of weak ECs. There are 28 strong (21.7%) and 15 super (11.6%) ECs, much larger than those over the Japan Sea. The maximum MDR over the Japan Sea (3.06 MBer) is smaller than that over the Kuroshio/Kuroshio Extension (3.59 MBer). The averaged MDR for ECs over the Kuroshio/Kuroshio Extension is 1.64 MBer, larger than 1.37 MBer for ECs over the Japan Sea, indicating that ECs over the Kuroshio/Kuroshio Extension generally develop more rapidly.

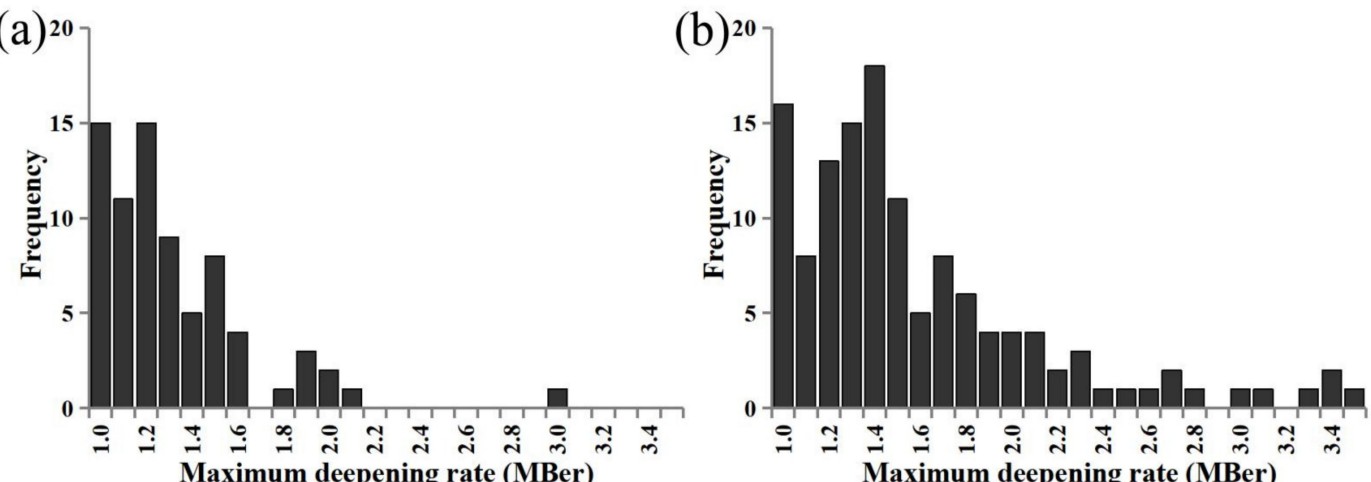

**Figure 3.** Frequency of the MDR for ECs over (**a**) the Japan Sea and (**b**) Kuroshio/Kuroshio Extension.

### 3.1.3. Frequency of Minimum Central SLP

Figure 4 shows the frequency of the minimum central SLP with 5 hPa intervals for ECs over the Japan Sea and Kuroshio/Kuroshio Extension. The frequency of the minimum central SLP peaks at 985–980 hPa for ECs over the Japan Sea and at 975–970 hPa for ECs over the Kuroshio/Kuroshio Extension. The frequency of the minimum central SLP for ECs over the Japan Sea decreases sharply from 975–970 hPa to 970–965 hPa, for ECs over the Kuroshio/Kuroshio Extension, and increases gradually from 1000–995 hPa to 975–970 hPa and decreases sharply from 965–960 hPa to 950–945 hPa. Over the Japan Sea, 78.7% (59) ECs have minimum central SLP between 995–975 hPa and 56.6% (73) ECs over the Kuroshio/Kuroshio Extension have minimum central SLP between 980–960 hPa. The minimum of the central SLP is 957.3 hPa for ECs over the Japan Sea and 935.6 hPa over the Kuroshio/Kuroshio Extension. The averaged minimum central SLP for ECs over the Japan Sea is 981.9 hPa, 7.8 hPa larger than 970.5 hPa for ECs over the Kuroshio/Kuroshio Extension. These results indicate that the minimum central SLP for ECs over the Kuroshio/ Kuroshio Extension is generally lower than that over the Japan Sea, implying that ECs over the Kuroshio/Kuroshio Extension can develop much more strongly.

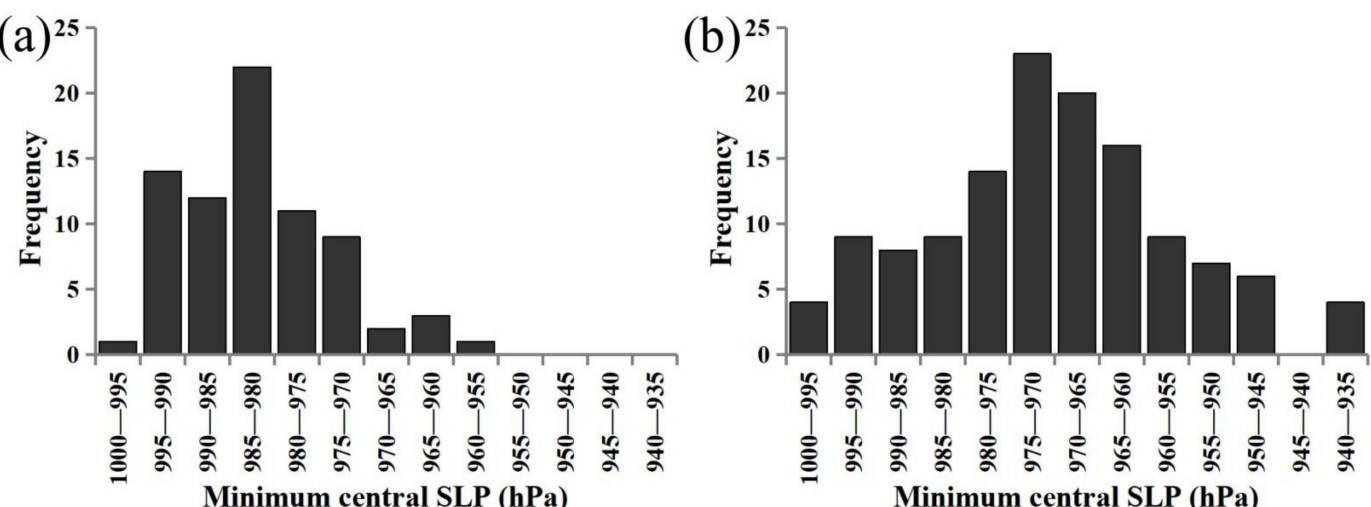

**Figure 4.** Frequency of minimum central SLP for ECs over (**a**) the Japan Sea and (**b**) Kuroshio/ Kuroshio Extension.

### 3.1.4. Frequency of Explosive-Developing Lifetime

Figure 5 depicts the frequency of the explosive-developing lifetime (measured in hours) for ECs over the Japan Sea and Kuroshio/Kuroshio Extension. The explosive-developing lifetime is defined as the duration with a deepening rate larger than or equal to 1 MBer. The frequency of the explosive-developing lifetime for ECs over the Japan Sea peaks at 3 h and 9 h, and decreases rapidly after 12 h. For ECs over the Kuroshio/Kuroshio Extension, the frequency has a maximum at 15 h, and decreases rapidly from 15 h to 18 h and then slowly from 18 h to 30 h. All ECs over the Japan Sea and 82.9% (107) ECs over the Kuroshio/Kuroshio Extension have explosive-developing lifetime shorter than 24 h. The maximum of explosive-developing lifetime for ECs over the Japan Sea is 21 h, shorter than 48 h for ECs over the Kuroshio/Kuroshio Extension. The averaged explosive-developing lifetime for ECs over the Japan Sea is 7.7 h, also shorter than 13 h for ECs over the Kuroshio/Kuroshio Extension. Therefore, the explosive-developing lifetime for ECs over the Kuroshio/Kuroshio Extension is distinctly longer than that over the Japan Sea.

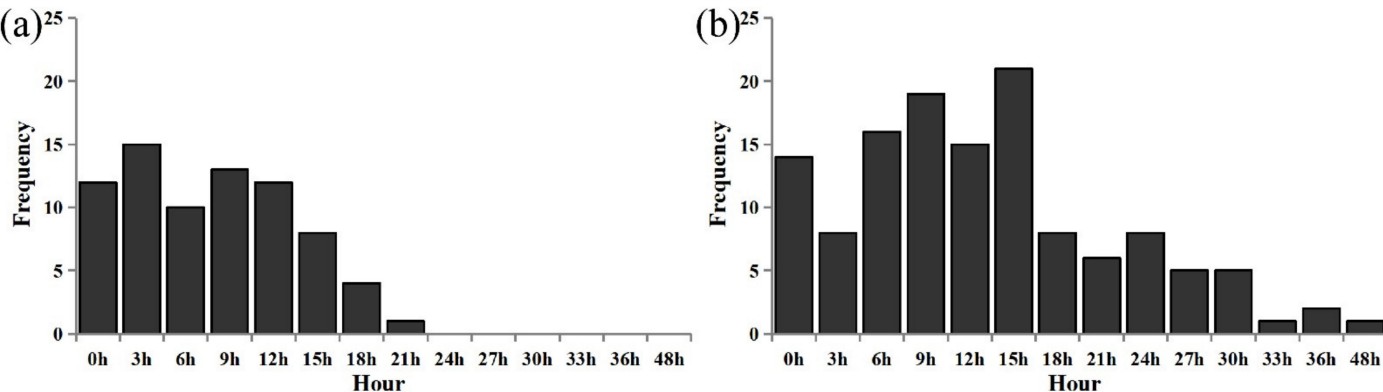

**Figure 5.** Frequency of the explosive-developing lifetime (in hours) for ECs over (**a**) the Japan Sea and (**b**) Kuroshio/Kuroshio Extension.

### 3.1.5. Moving Track

Figure 6 illustrates moving tracks for various intensity of ECs over the Japan Sea and Kuroshio/Kuroshio Extension. ECs over the Japan Sea usually form over continental Northeast Asia, eastern China and the Japan Sea. Moving tracks of ECs formed over eastern China and the Japan Sea generally have a northeastward direction, whereas those formed over the continental Northeast Asia usually move eastward. They mostly dissipate over the Okhotsk Sea, except that some ECs dissipate over the northeastern Pacific. Moving tracks of weak and moderate ECs over the Japan Sea are similar; the strong and super ECs over the Japan Sea usually form over eastern China and the Japan Sea, and move in a northeastward direction. Moving tracks of ECs over the Kuroshio/Kuroshio Extension are typically in a northeastward direction and travel along the Kuroshio/Kuroshio Extension. They mainly form over the East China Sea, except that some weak ECs originate over the Japan Sea. Most weak ECs over the Kuroshio/Kuroshio Extension form over the sea to the south and southeast of the Japan Islands. As the MDR of ECs enhances, ECs formed over the East China Sea tend to increase and can develop more rapidly. Moreover, the explosive-developing tracks (the track with a deepening rate larger than or equal to 1 MBer, red line in Figure 6) of the weak and moderate ECs generally distribute over the sea to the east of the Japan Islands, while strong and super ECs begin to develop explosively over the sea to the south of the Japan Islands with longer explosive-developing tracks. They generally disappear over the northwestern Pacific, except for some over the Okhotsk Sea. Moving tracks for ECs over the Kuroshio/Kuroshio Extension are usually longer than those over the Japan Sea.

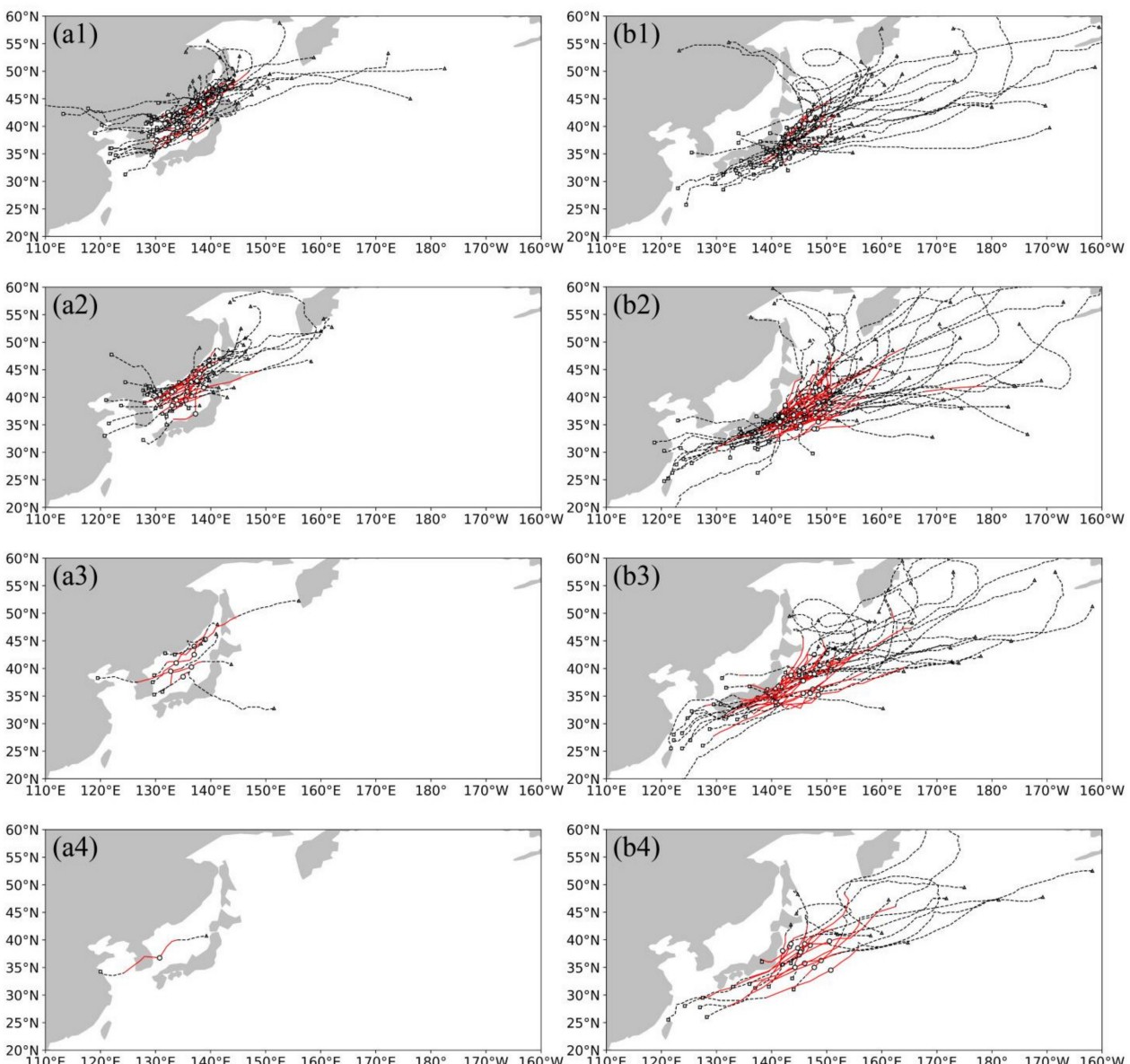

**Figure 6.** Moving tracks of (**a1**,**b1**) weak, (**a2**,**b2**) moderate, (**a3**,**b3**) strong and (**a4**,**b4**) super ECs over the Japan Sea (left column) and Kuroshio/Kuroshio Extension (right column). "□" is the EC center at the formation moment, "○" is the EC center at the maximum-deepening-rate moment, "Δ" is the EC center at the ending moment. The dashed line indicates the track with a deepening rate smaller than 1 MBer, while the red-solid line indicates the track with a deepening rate larger than or equal to 1 MBer.

### 3.2. Composite Synoptic-Scale Environmental Conditions

3.2.1. Low-Level Baroclinicity

ECs generally have strong baroclinicity at low level during their rapid development [14,20,22,38,39]. The baroclinicity is represented by the horizontal gradient of the potential temperature. The composite fields of 850 Pa (Figure 7a1,b1) and 500 hPa (Figure 7a2,b2) potential temperature and baroclinicity for ECs over the Japan Sea (Figure 7a1,a2) and Kuroshio/Kuroshio Extension (Figure 7b1,b2) are shown in Figure 7. The trough and ridge of the potential temperature are located in the upstream and downstream of both EC areas, respectively, at 850 hPa and 500 hPa, where the amplitude of potential temperature disturbance for ECs over the Japan Sea is deeper than that over the Kuroshio/Kuroshio Extension. The baroclinicity is located between the trough and

ridge of the potential temperature and shows a southwest-northeast orientation. The Japan Sea is dominated by strong baroclinicity extended along the west coast, whereas the relatively weaker baroclinicity for ECs over the Kuroshio/Kuroshio Extension distributes over the east coast of the Japan Islands. The baroclinicity at 850 hPa is weaker than that at 500 hPa over both areas and the difference is more significant for ECs over the Kuroshio/Kuroshio Extension.

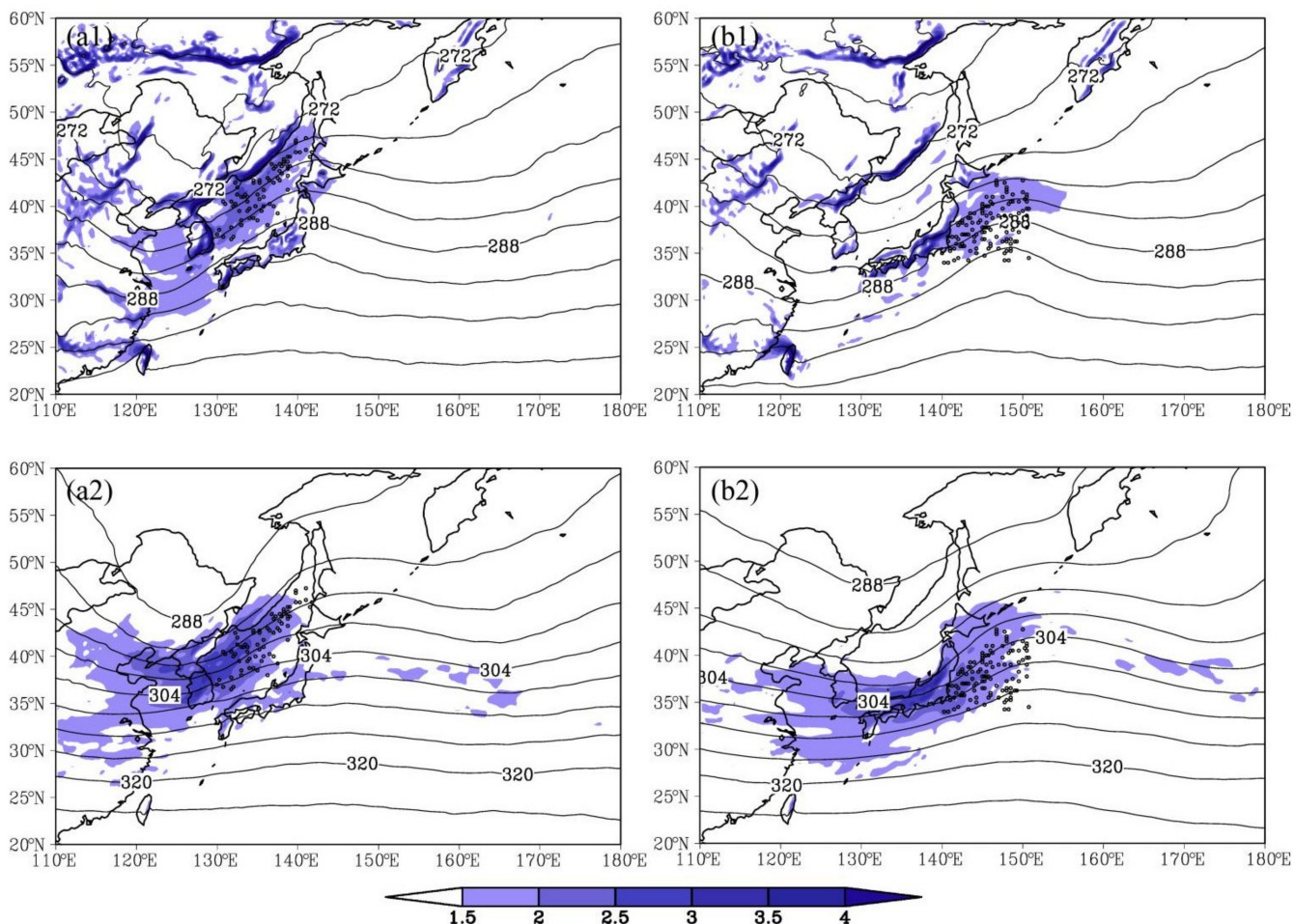

**Figure 7.** Composite fields of potential temperature (solid line, 4 K intervals) and baroclinicity (shaded, $0.5 \times 10^{-5}$ K m$^{-1}$ intervals) for ECs at maximum-deepening-rate moment over (**a1,a2**) the Japan Sea and (**b1,b2**) Kuroshio/Kuroshio Extension at (**a1,b1**) 850 hPa and (**a2,b2**) 500 hPa. "∘" is the EC center at maximum-deepening-rate moment.

Figure 8 shows the corresponding anomaly fields of the potential temperature and baroclinicity for these two areas, respectively. The significant negative and positive anomalies of potential temperature superimpose over the trough and ridge of the potential temperature, strengthening its amplitude (Figure 7). The RAS of potential temperature is generally smaller than −1 or larger than 1 over the negative and positive anomaly areas for ECs over the Japan Sea, and it is larger than 1 over the positive anomaly areas for ECs over the Kuroshio/Kuroshio Extension. The negative anomaly of potential temperature for ECs over the Japan Sea (Figure 8a1,a2) is significantly stronger than that over the Kuroshio/Kuroshio Extension (Figure 8b1,b2), but the positive anomaly of potential temperature for ECs over the Kuroshio/Kuroshio Extension is slightly stronger than that over the Japan Sea, which indicates that the baroclinicity for ECs over the Japan Sea mainly results from the intrusion of cold air and that for ECs over the Kuroshio/Kuroshio Extension comes from the heating of the underlying warm current. The gradient of negative and positive anomalies of the

potential temperature over the Japan Sea is larger than that over the Kuroshio/Kuroshio Extension, resulting in the anomaly of baroclinicity over the Japan Sea being stronger than that over the Kuroshio/Kuroshio Extension, while the RAS of the baroclinicity is smaller than 1. The baroclinicity anomaly to the north of the Kuroshio/Kuroshio Extension is probably related to the bant-benk front, a significant feature for ECs over the warm current [26,40]. These results indicate that the baroclinicity anomaly favors the development of ECs over the Japan Sea and Kuroshio/Kuroshio Extension, and anomalies over the Japan Sea are generally stronger than those over the Kuroshio/Kuroshio Extension.

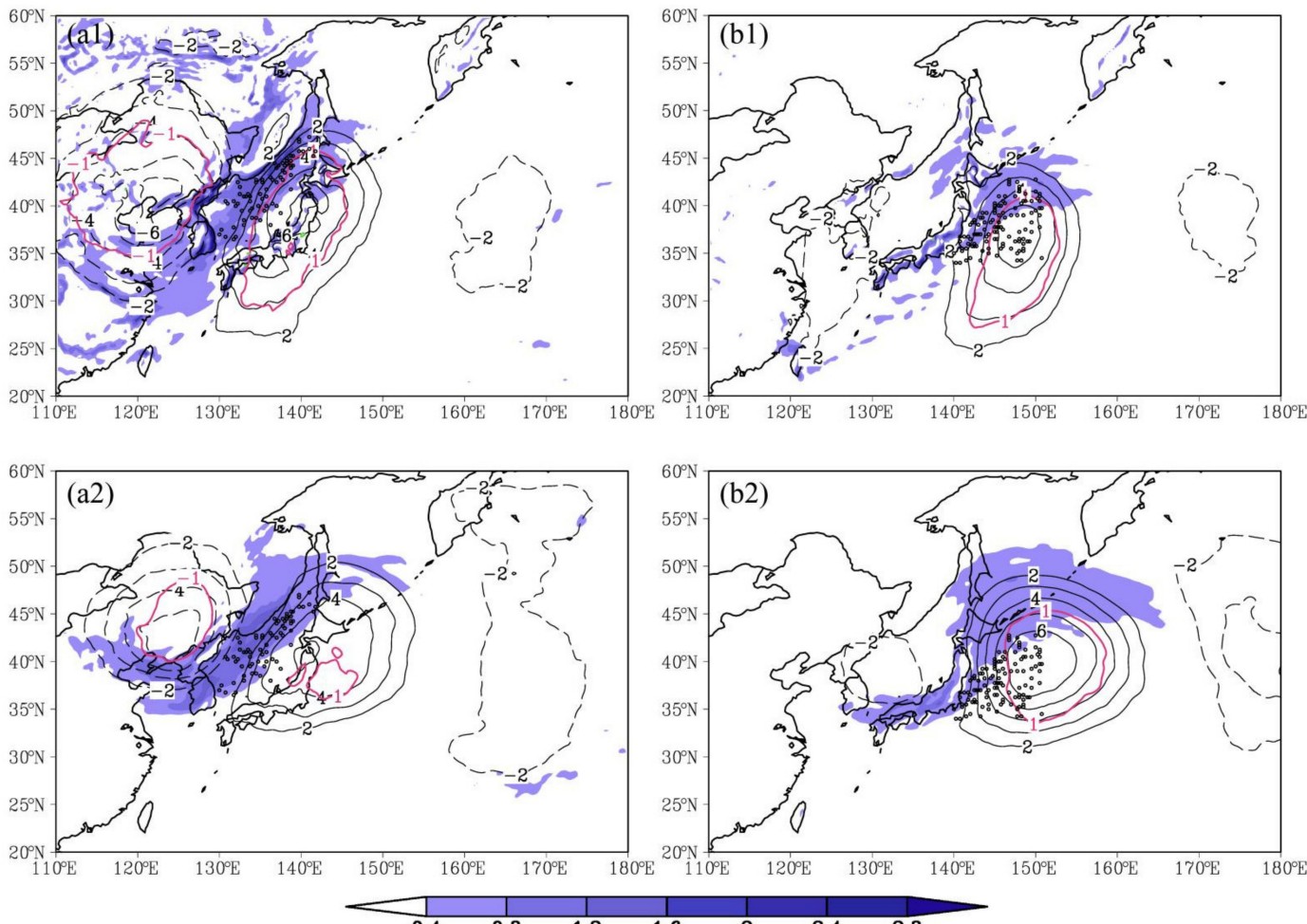

**Figure 8.** Anomaly fields of composite potential temperature (black line, 1 K intervals) and baroclinicity (shaded, $0.4 \times 10^{-5}$ K m$^{-1}$ intervals) for ECs at maximum-deepening-rate moment over (**a1,a2**) the Japan Sea and (**b1,b2**) Kuroshio/Kuroshio Extension at (**a1,b1**) 850 hPa and (**a2,b2**) 500 hPa. The red line indicates the RAS of potential temperature equal to 1 or −1.

Previous studies indicated that ECs were fundamentally driven by baroclinicity [34,41,42]. Yoshida and Asuma [8] pointed out that the favorable atmospheric conditions for EC development over the Japan and Okhotsk Sea and the northwestern Pacific are closely related to the presence and extension of the cold air mass from the Asian continent. The East Asian winter monsoon transports this cold air mass to the Japan Sea and Kuroshio/Kuroshio Extension in the cold season, whereas the cold air mass will be warmed and will experience modification when passing through the sea over the upstream of the Japan Islands, resulting in relatively weaker baroclinicity over the Kuroshio/Kuroshio Extension than that over the Japan Sea. The baroclinicity over these two areas is strong and favors the frequent occurrence of ECs. However, the key factor affecting baroclinicity over

the Japan Sea is the intrusion of cold air and that over the Kuroshio/Kuroshio Extension is the strong heating of the warm current.

### 3.2.2. Low-Level Water Vapor

As the latent heat release is considered to be an important factor for rapid EC development [14–16,24,43], low-level water vapor, which can rise and lead to the latent heat release, is shown in Figure 9. The tongue of the specific humidity extends northeastward in the downstream of ECs over the Japan Sea (Figure 9a1,a2) and Kuroshio/Kuroshio Extension (Figure 9b1,b2) at 925 hPa (Figure 9a1,b1) and 850 hPa (Figure 9a2,b2). The horizontal winds are southwesterly and southerly low-level jet stream over the specific humidity tongue over these two areas, respectively, with the weaker low-level jet stream over the Japan Sea than that over the Kuroshio/Kuroshio Extension. The stronger tongue of specific humidity associated with the stronger southerly low-level jet stream over the Kuroshio/Kuroshio Extension results in larger water vapor convergence than that over the Japan Sea at 925 hPa and 850 hPa. In addition, the areas of larger water vapor convergence over the Japan Sea can be found to the northeast and southwest of Japan Islands at 925 hPa and 850 hPa, and the lower water vapor convergence can be identified over the Japan Islands, which implies that the Japan Islands, with high elevation, probably block some of the moisture.

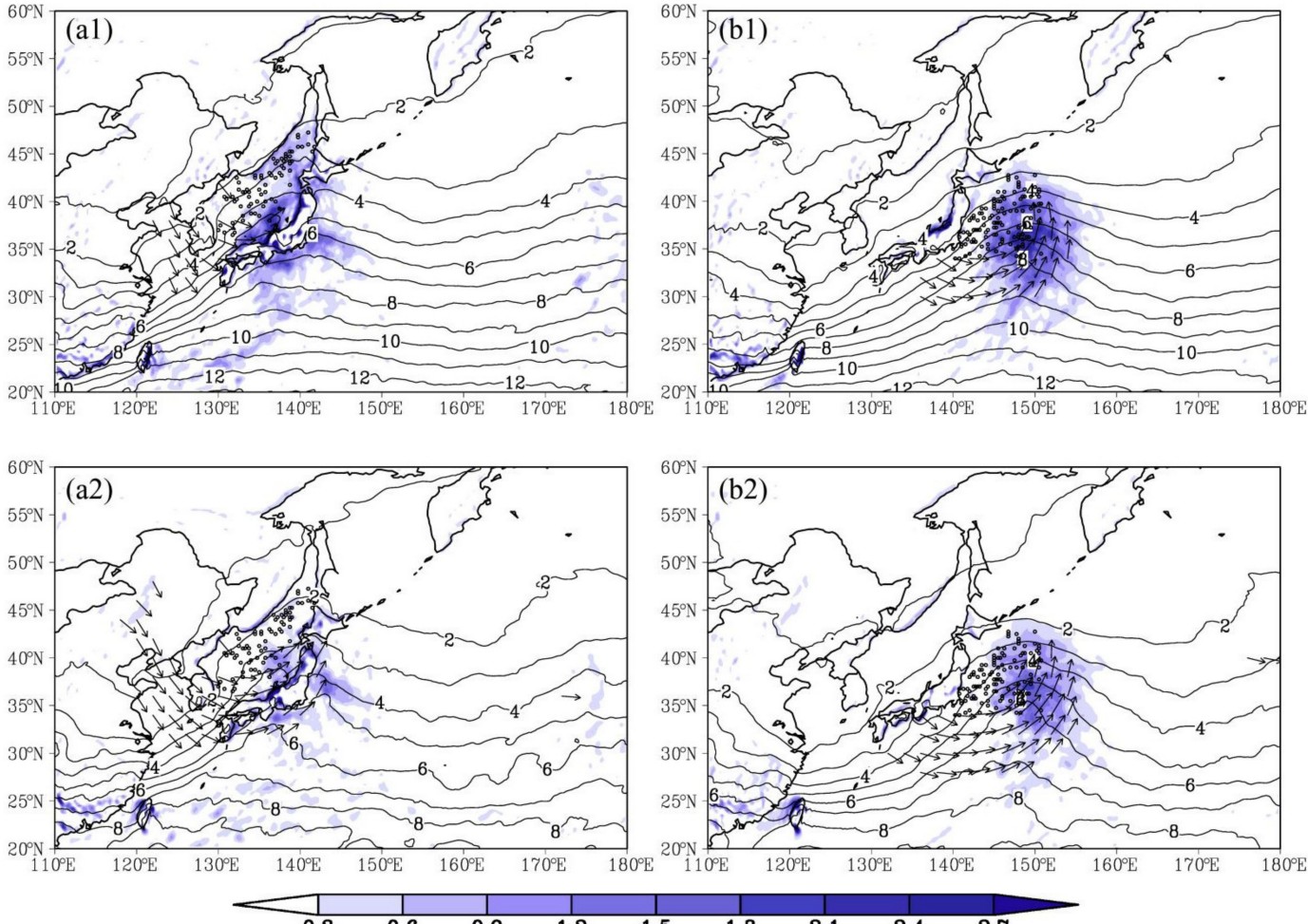

**Figure 9.** Composite fields of horizontal wind vector (arrow, $\geq 12$ m s$^{-1}$), specific humidity (solid line, 1 g kg$^{-1}$ interval) and water vapor convergence (shaded, $0.3 \times 10^{-4}$ g kg$^{-1}$ s$^{-1}$ intervals) for ECs at maximum-deepening-rate moment over (**a1,a2**) the Japan Sea and (**b1,b2**) Kuroshio/Kuroshio Extension at (**a1,b1**) 925 hPa and (**a2,b2**) 850 hPa. "○" is the EC center at the maximum-deepening-rate moment.

The anomaly fields of the horizontal wind vector, specific humidity and water vapor convergence for ECs over the Japan Sea and Kuroshio/Kuroshio Extension are shown in Figure 10. Wind anomalies for ECs over the Japan Sea (Figure 10a1,a2) and Kuroshio/ Kuroshio Extension (Figure 10b1,b2) are cyclonic and the intensity at 925 hPa (Figure 10a1,b1) is generally stronger than that at 850 hPa (Figure 10a2,b2). The southerly wind anomaly for ECs over the Japan Sea is weaker than the southeasterly wind anomaly over the Kuroshio/Kuroshio Extension in the east of the EC areas. For ECs over the Kuroshio/ Kuroshio Extension, the positive anomaly of specific humidity dominates EC areas and is centered in the southeast, while that to the north and southeast for ECs over the Japan Sea is weaker. The anomaly of specific humidity is generally larger than its standard error for these two EC areas (RAS ≥ 1). The strong southeasterly wind anomaly for ECs over the Kuroshio/Kuroshio Extension associated with larger specific humidity transports abundant moist air, leading to the larger water vapor convergence in the east. In addition, two larger anomaly centers of water vapor convergence for ECs over the Japan Sea can be found to the northwest and southeast of the Japan Islands and the lower water vapor convergence can be identified there, consistent with the mean composite fields.

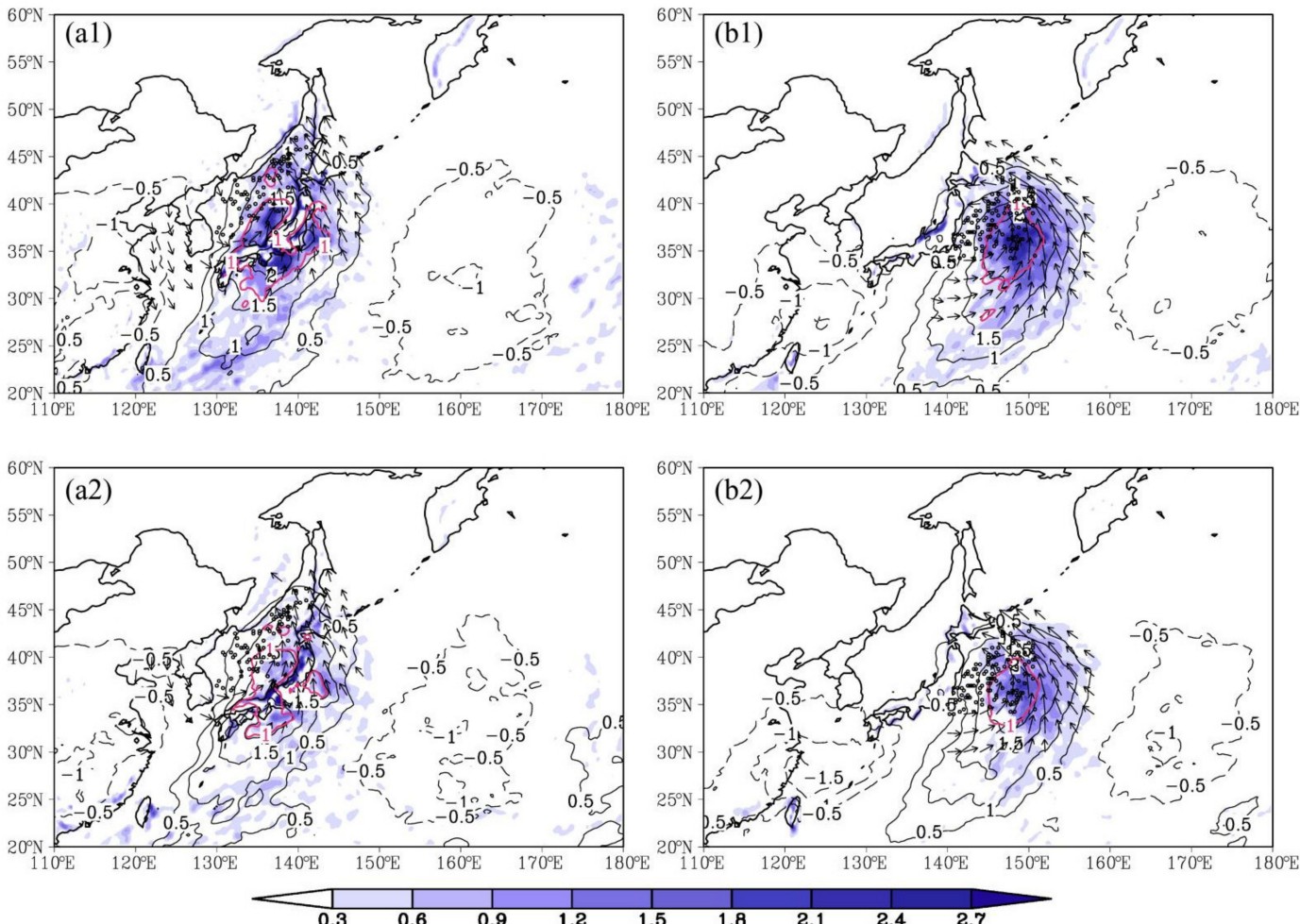

**Figure 10.** Anomaly fields of composite horizontal wind vector (arrow, $\geq 8$ m s$^{-1}$), specific humidity (black line, 0.5 g kg$^{-1}$ intervals) and water vapor convergence (shaded, $0.3 \times 10^{-4}$ g kg$^{-1}$ s$^{-1}$ intervals) for ECs at maximum-deepening-rate moment over (**a1,a2**) the Japan Sea and (**b1,b2**) Kuroshio/Kuroshio Extension at (**a1,b1**) 925 hPa and (**a2,b2**) 850 hPa. The red line indicates the RAS of potential temperature equal to 1 or −1.

ECs over the Kuroshio/Kuroshio Extension have abundant water vapor evaporated from the underlying warm ocean surface and abundant latent heat can be released from condensation, contributing significantly to the rapid EC development over this area, as pointed out in other case studies [14–16,24,43,44], whereas the water vapor convergence for ECs over the Japan Sea is weaker, because of the weaker southwesterly wind and the prevention of the moist air transportation from the Kuroshio/Kuroshio Extension by the Japan Islands. Moreover, the cold front of ECs over the Japan Sea usually intruded to the Kuroshio, and the northward moisture transport was consumed on the south coast of Japan by cold front or cyclone [45], which may be another possible mechanism. Therefore, the low-level water vapor contributes more significantly to ECs over the Kuroshio/Kuroshio Extension than those over the Japan Sea [14,35].

### 3.2.3. Mid-Upper Level Cyclonic Vorticity

The mid-upper tropospheric environmental condition also affects EC dynamics [29,41]. Figure 11 shows the composite fields of the geopotential height and vorticity for ECs over the Japan Sea (Figure 11a1,a2) and Kuroshio/Kuroshio Extension (Figure 11b1,b2) at 500 hPa (Figure 11a1,b1) and 300 hPa (Figure 11a2,b2). The pattern of the geopotential height at middle level (500) and upper level (300hPa) is similar, but shows significant difference in areas. The trough of the geopotential height over the Japan Sea is deeper, whereas it is shallower over the Kuroshio/Kuroshio Extension. The strong cyclonic vorticity in the trough dominates the upstream of ECs over these two areas, because of the cyclonic-vorticity advection by southwesterly winds, favoring rapid EC development. The cyclonic vorticity at 300 hPa is generally stronger than that at 500 hPa. The cyclonic vorticity over the Japan Sea at 500 hPa and 300 hPa are both stronger than those over the Kuroshio/Kuroshio Extension. Thus, ECs over the Japan Sea have a deeper synoptic-scale trough of geopotential height associated with stronger cyclonic vorticity, whereas ECs over the Kuroshio/Kuroshio Extension have shallower synoptic-scale trough of geopotential height associated with relatively weaker cyclonic vorticity.

The mid-upper tropospheric characteristics for ECs over the Japan Sea and Kuroshio/Kuroshio Extension are associated with their synoptic-scale disturbances (Figure 12). The significant negative and positive anomalies of geopotential height are located over the trough and ridge of their geopotential height over both Japan Sea (Figure 12a1,a2) and Kuroshio/Kuroshio Extension (Figure 12b1,b2) at 500 hPa (Figure 12a1,b1) and 300 hPa (Figure 12a2,b2). The negative anomaly of the geopotential height over Japan Sea is obviously stronger than that over the Kuroshio/Kuroshio Extension, increasing the amplitude of the trough. The negative and positive anomalies of the geopotential height are generally larger than its standard deviation for EC over the Japan Sea (RAS $\leq -1$ or $\geq 1$), while only the positive anomaly of geopotential height is larger than its standard error for EC over the Kuroshio/Kuroshio Extension (RAS $\geq 1$). The anomaly of strong cyclonic vorticity distributes over the upstream of two EC areas. Meanwhile the anomaly of cyclonic vorticity for ECs over the Japan Sea is significantly stronger than that over the Kuroshio/Kuroshio Extension, similar to the characteristics of the cyclonic vorticity in Figure 11. The anomaly of cyclonic vorticity at 300 hPa for ECs over the Japan Sea is larger than its standard deviation (RAS $\geq 1$). These anomalies of geopotential height and cyclonic vorticity produce favorable conditions for EC development over these two areas, especially over the Japan Sea.

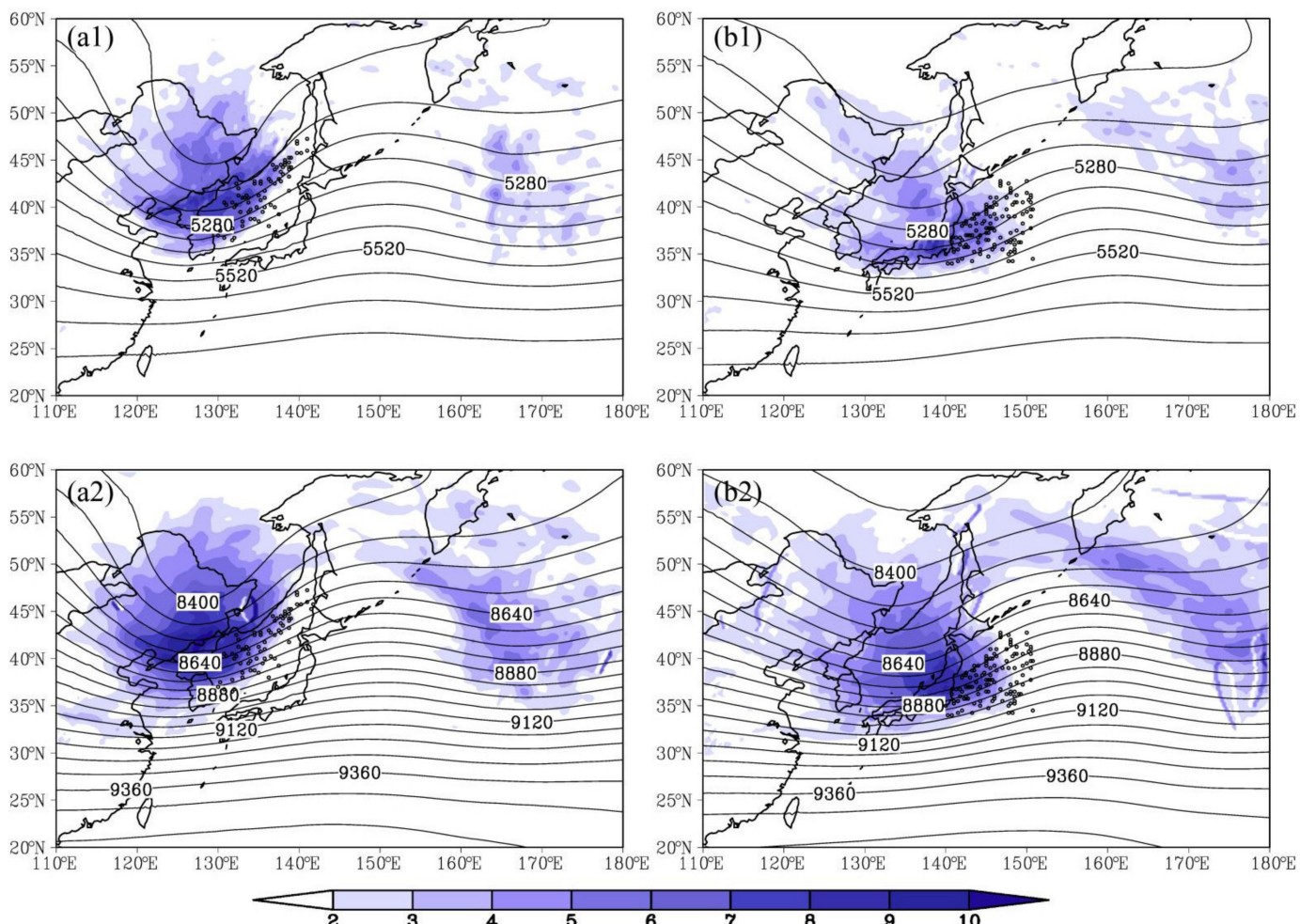

**Figure 11.** Composite fields of geopotential height (solid line, 40 pgm intervals) and relative vorticity (shaded, $1 \times 10^{-5}$ s$^{-2}$ intervals) for ECs at the maximum-deepening-rate moment over (**a1,a2**) the Japan Sea and (**b1,b2**) Kuroshio/Kuroshio Extension at (**a1,b1**) 500 hPa and (**a2,b2**) 300 hPa. "∘" is the EC center at the maximum-deepening-rate moment.

Sanders and Gyakum [1] suggested that ECs usually occurred in the downstream of the 500 hPa trough. The wave trough in the mid-upper troposphere favors the rapid EC development [21,29,41,46]. The geopotential height off the east Asian coast generally shows a trough associated with the cyclonic vorticity in the cold season [2], leading to the cyclonic-vorticity advection in the downstream and providing favorable environmental condition for frequent EC occurrence. The composite and corresponding anomaly fields of the geopotential height and cyclonic vorticity for ECs over the Japan Sea are stronger than those over the Kuroshio/Kuroshio Extension, implying that they contribute more to the rapid EC development over the Japan Sea.

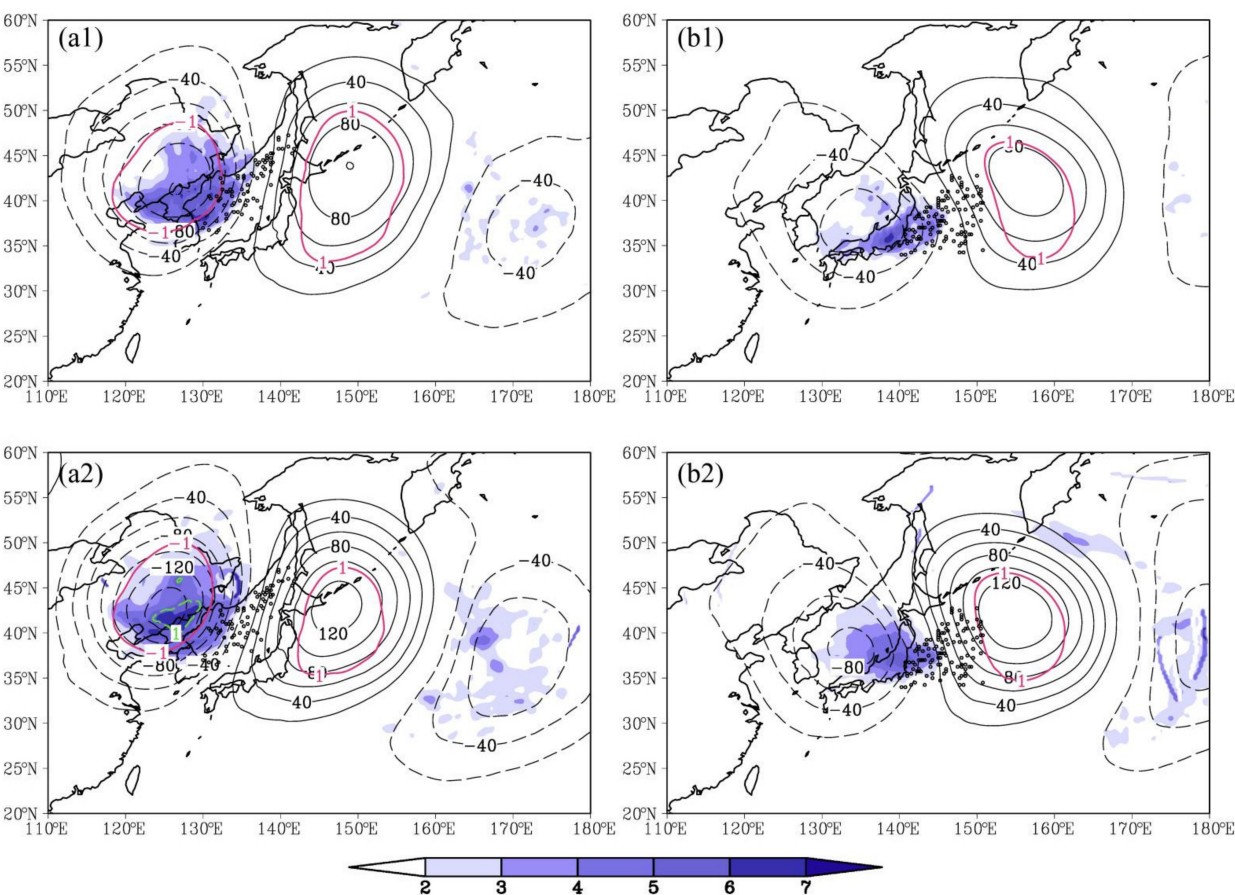

**Figure 12.** Anomaly fields of composite geopotential height (black line, 20 pgm intervals) and relative vorticity (shaded, $1 \times 10^{-5}$ s$^{-2}$ intervals) for ECs at the maximum-deepening-rate moment over (**a1,a2**) the Japan Sea and (**b1,b2**) Kuroshio/Kuroshio Extension at (**a1,b1**) 500 hPa and (**a2,b2**) 300 hPa. The red and green lines indicate the RAS of geopotential height and relative vorticity equal to 1 or $-1$.

3.2.4. Upper-Level Jet Stream and Potential Vorticity

As some studies have emphasized that upper-level forcing contributes significantly to the rapid EC development [8,21,34,47], the composite fields of 300 hPa jet stream and PV for ECs over the Japan Sea and Kuroshio/Kuroshio Extension are presented in Figure 13. The upper-level jet stream for ECs over the Japan Sea extends from the east-central China to the central Japan Islands with a southwest-northeast orientation and is centered over the Japan Islands with wind speed greater than 55 m s$^{-1}$. Compared with the upper-level jet stream over the Japan Sea, the upper-level jet stream over the Kuroshio/Kuroshio Extension extends more zonally and shows stronger wind speed, greater than 60 m s$^{-1}$, to the south of the Japan Islands. ECs over the Japan Sea and Kuroshio/Kuroshio Extension are located in the downstream of their troughs and the left of the jet stream exit. The strong PV dominates the upstream of ECs over these two areas. The PV center of 5.5 PVU over the Japan Sea is much stronger than the center of 4 PVU over the Kuroshio/Kuroshio Extension.

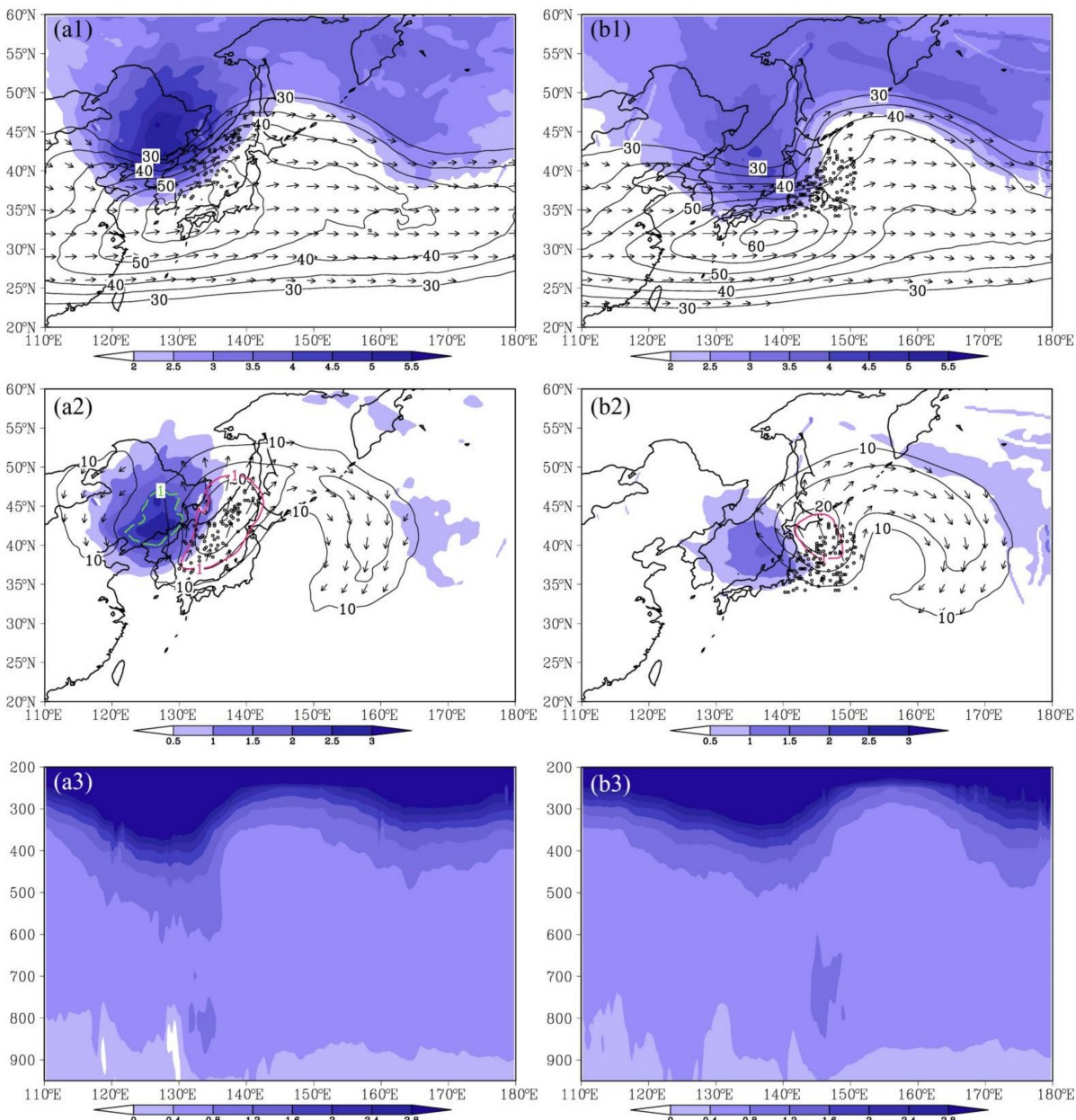

**Figure 13.** Composite fields of 300 hPa jet stream (solid line, 5 m s$^{-1}$ intervals), horizontal wind vector (arrow, $\geq$30 m s$^{-1}$) and PV (shaded, 0.5 PVU intervals) for ECs at maximum-deepening-rate moment over (**a1**) the Japan Sea and (**b1**) Kuroshio/Kuroshio Extension. The corresponding anomaly fields are in (**a2,b2**), The red and green lines indicate the RAS of geopotential height and relative vorticity equal to 1 or $-1$. Note the horizontal wind vector is $\geq$ 10 m s$^{-1}$. (**a3,b3**) are vertical cross sections of composite PV (0.4 PVU intervals) along 42$^\circ$ N and 40$^\circ$ N for ECs over the Japan Sea and Kuroshio/Kuroshio Extension, respectively. "○" is the EC center at the maximum-deepening-rate moment.

Wind anomalies shown in Figure 13a2,b2 are anticyclonic over two EC areas. The southwesterly anomaly (with wind speed greater than 25 m s$^{-1}$) for ECs over the Japan Sea is over the central Japan Sea, whereas that for ECs over the Kuroshio/Kuroshio Extension flows southeasterly and is centered to the north (with relatively weaker speed greater than 20 m s$^{-1}$). The wind anomalies are generally larger than its standard deviation for these two EC areas (RAS $\geq$ 1). The large PV area for ECs over these two areas distribute in the upstream and that for ECs over the Japan Sea is much stronger. The large PV area at 300 hPa for ECs over Japan Sea is larger than its standard deviation (RAS $\geq$ 1). The vertical cross

sections of the composite PV for ECs over Japan Sea (crossing the PV center at about 42° N) and Kuroshio/Kuroshio Extension (crossing the PV center at about 40° N) are shown in Figure 13a3,b3, respectively. An upper-level large PV area appears in the upstream of both EC areas. The PV of 0.8 PVU extends further downwards below 600 hPa for ECs over the Japan Sea than that at about 500 hPa for ECs over the Kuroshio/Kuroshio Extension. The low-level large PV area of 0.8 PVU centers near 800 hPa for ECs over the Japan Sea, lower than that near 700 hPa for ECs over the Kuroshio/Kuroshio Extension. The means the low-level PV patches (≥0.8 PVU) over both EC areas are statistically significantly higher than the surrounding area (900–700 hPa) mean at 95% confidence level using the Student's *t*-test.

Rapid EC development is usually associated with the strong upper-level jet stream [21,34,47]. The divergence and cyclonic-vorticity advection usually appear at the exit of the jet stream [47–49], which is closely related to the surface cyclogenesis [34,50]. The East Asian subtropical westerly jet stream is generally located over the south of the Japan Islands in the cold season [2], which results in the strong upper-level jet stream, favoring frequent EC occurrences over the Kuroshio/Kuroshio Extension. The vertical cross section of the composite PV showed a large area in upper and low levels. The interaction between upper and low-level PV could lead to an intensification of the cyclogenesis [31], which was a typical characteristic of strong extratropical cyclones [5,22,32,33,51].

## 4. Discussion and Conclusions

ECs during 15-year (2000–2015) cold-seasons (October–April) are identified and tracked using an objective method and high resolution ERA5 atmospheric reanalysis. Results show that the Japan Sea and Kuroshio/Kuroshio Extension have the highest EC occurrences over the northern Pacific (Figure 1). The statistical characteristics and composite synoptic-scale environmental conditions for ECs over these two areas are examined and compared. There are 75 ECs over the Japan Sea and 129 ECs over the Kuroshio/Kuroshio Extension in this analysis. Statistical features, including the minimum central SLP, explosive-developing lifetime and track, which were seldom analyzed in previous studies [8,14], have been examined in order to gain further understanding of EC characteristics over these two areas.

The EC frequency shows evident seasonal variations over the Japan Sea and Kuroshio/ Kuroshio Extension. ECs over the Japan Sea frequently occur in late autumn (November) and early winter (December), while ECs over the Kuroshio/Kuroshio Extension mainly occur in winter (December, January and February) and early spring (March). The frequency of MDR generally decreases with the increase of MDR, while the MDRs for ECs over the Kuroshio/Kuroshio Extension are generally larger than that over the Japan Sea, indicating that the ECs over the Kuroshio/Kuroshio Extension develop more rapidly. The minimum central SLP for ECs over the Kuroshio/Kuroshio Extension is generally lower than that over the Japan Sea, implying that ECs over the Kuroshio/Kuroshio Extension can develop much more strongly. The explosive-developing lifetime for ECs over the Kuroshio/Kuroshio Extension is distinctly longer than that over the Japan Sea. Moving tracks for ECs over the Japan Sea are in a northeastward or eastward direction, and those over the Kuroshio/Kuroshio Extension are typically in a northeastward direction. The strong and super ECs over the Japan Sea usually form over East China and the Japan Sea. ECs over the Kuroshio/Kuroshio Extension formed over East China Sea generally can develop more rapidly. Moreover, explosive-developing tracks of weak and moderate ECs generally distribute over the sea to the east of the Japan Islands, while strong and super ECs begin to develop explosively over the sea to the south of the Japan Islands with a longer explosive-developing tracks.

The composite and anomaly fields show that ECs over the Japan Sea and Kuroshio/ Kuroshio Extension are characterized by different synoptic-scale atmospheric environmental conditions. ECs over the Japan Sea have higher low-level baroclinicity (Figure 7a1,a2), poorer low-level water vapor convergence (Figure 9a1,a2), larger mid-upper level cyclonic

vorticity (Figure 11a1,a2) and weaker upper-level jet stream (Figure 13a1), compared with those over the Kuroshio/Kuroshio Extension (Figure 7b1,b2, Figure 9b1,b2, Figure 11b1,b2, and Figure 13b1). For ECs over these two areas, the baroclinicity at 850 hPa is weaker than that at 500 hPa. The key factor contributing to the baroclinicity for ECs over the Japan Sea is the cold air intrusion and that over the Kuroshio/Kuroshio Extension is the strong heating of the warm current. The baroclinicity anomaly to the north of the Kuroshio/Kuroshio Extension is probably related to the bant-benk front, a significant feature for ECs over the warm current [39,40]. These findings are different from previous studies [8]. The warm current provides abundant moist air for ECs over the Kuroshio/Kuroshio Extension, while the air over the Japan Sea is relatively dry. Lower water vapor convergence can be identified over Japan Islands, because their high elevation can block the moist air, and the northward moisture transport can be consumed in the south coast of Japan by cold front or cyclone [45]. The trough of East Asia associated with the cyclonic vorticity favors the generation of the larger mid-upper level cyclonic-vorticity advection for ECs over the Japan Sea than that over the Kuroshio/Kuroshio Extension. The East Asian subtropical westerly jet stream is generally located over the south of the Japan Islands in the cold season and contributes more to EC development over the Kuroshio/Kuroshio Extension. The large PV area is distributed in the upper and low levels over both EC areas and extends further downwards for ECs over the Japan Sea.

ECs over the northwestern Pacific have been investigated by previous studies, particularly by Yoshida and Asuma [8]. The current study uses high resolution and longer period ERA5 reanalysis and more ECs are identified with an even higher EC threshold than Yoshida and Asuma [8]. The statistical features of the monthly frequency and MDR show differences to the results of Yoshida and Asuma [8]. More statistical features such as the minimum central SLP, explosive-developing lifetime and explosive-developing track are also examined in this study. The baroclinicity for ECs over the Japan Sea is stronger than that over the Kuroshio/Kuroshio Extension, consistent generally with the results of Yoshida and Asuma [8], while key factors contributing to the baroclinicity over these two EC areas are further analyzed, and the main factor over the Japan Sea is the cold air intrusion and that over the Kuroshio/Kuroshio Extension is the strong heating of the warm current. Kuwao-Yoshida and Asuma [14] indicated that explosive cyclones over the Japan Sea have less moisture contribution than in the case study. The possible mechanism for the weak moisture contribution is further discussed in our composite analysis. In addition, the influences of the vorticity and PV on rapid EC development are also investigated by the composite analysis. The synoptic-scale environmental conditions for different cases should be further investigated and compared to give a comprehensive understanding of EC development in future work.

**Author Contributions:** Conceptualization, S.Z. and R.T.; methodology, S.Z. and C.L.; software, Y.Z.; validation, Y.X.; formal analysis, S.Z., J.X. and J.L.; data curation, Y.Z. and X.G.; writing—original draft preparation, S.Z., G.F. and R.T.; writing—review and editing S.Z., J.L., G.F., Y.X. and C.L.; visualization, Y.Z. and J.X. All authors have read and agreed to the published version of the manuscript.

**Funding:** This research was jointly funded by Project of Enhancing School with Innovation of Guangdong Ocean University (230419106); The State Key Program of National Natural Science Foundation of China (42130605); National Natural Science Foundation of China (42075036, 41976200 and 41775042); Guangdong Basic and Applied Basic Research Foundation (2019B1515120018); National Key Research and Development Project (2018YFC1506902), Guangdong Ocean University PhD. Scientific Research Program (R19045); Scientific Research Start-up Grant of Guangdong Ocean University (R20001); Fujian Key Laboratory of Severe Weather (2021KFKT02).

**Data Availability Statement:** ECWMF ERA5 data can be downloaded from https://cds.climate.copernicus.eu/cdsapp#!/dataset/reanalysis-era5-single-levels-monthly-means?tab=form, accessed on 22 December 2021.

**Acknowledgments:** The authors would like to express their sincere thanks to funding organizations, and special thanks go to ECMWF for providing ERA5 data. We would like to thank the reviewers for their valuable suggestions and comments.

**Conflicts of Interest:** The authors declare no conflict of interest.

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
