# Peer review of "Statistical Characteristics and Composite Environmental Conditions of Explosive Cyclones over the Japan Sea and Kuroshio/Kuroshio Extension"

_atmosphere, doi:10.3390/atmos13010017_

Round 1
Reviewer 1 Report
I suggest it be accepted except one very minor point as below:
For the authors' response to point 2, MB to me is a bit confusing, since this is similar to millibar. Perhaps change to 'MBer'?
Reviewer 2 Report
Statistical differences in explosive cyclones between the Japan Sea and Kuroshio/its extension are interesting to me. I'm satisfied with the authors' responses to my review comments. Although I recommend the acceptance, the authors should consider the following minor comments.
L381, L383, L435, L437, L442, L482, L485, ... : Change lager to larger.
L491 to L493: I think that the low-level PV patches of >0.8 PVU are weak compared with the background PV around the patches in Fig. 13a3 & 13b3. Please comment or discuss whether the PV patches at 800-700 hPa are statistically significant, or not.
L687 (Ref. 40): Change the first names to the family names of the authors.
Author Response
Please see the attachment.

This manuscript is a resubmission of an earlier submission. The following is a list of the peer review reports and author responses from that submission.
Round 1
Reviewer 1 Report
This manuscript statistically examined the features of explosive cyclones over the Japan Sea and the Kuroshio/Kuroshio Extension. However, the most part of the results of this study reaffirm the findings of previous studies (e.g., Yoshida and Asuma 2004, Kuwao-Yoshida and Asuma 2008). Therefore, this manuscript cannot significantly contribute to deepening our understanding of the development of explosive cyclones over the East Asian region. Moreover, I have some serious concerns. Based on these reasons, my recommendation for this manuscript is reject. Specific comments are noted below.
Major comments:
Seasonal variability of frequency of explosive cyclones around the Japan Sea and the Kuroshio/Kuroshio Extension (Fig. 2) was well investigated by Yoshida and Asuma (2004). Statical features of MDR (Fig. 3) and tracks (Fig. 7) of cyclones are also investigated by Yoshida and Asuma (2004). These results of this study are consistent with Yoshida and Asuma (2004).
Section 3 studied the factors related to the development of explosive cyclones over the Japan Sea and the Kuroshio/Kuroshio Extension. The results indicated that baroclinicity and cyclonic-vorticity advection is important for the development of the Japan Sea cyclones, while the moist process leads to the rapid development the Kuroshio/Kuroshio Extension cyclones. These results are same as the conclusion of the statical investigations of Yoshida and Asuma (2004). The claim of Yoshida and Asuma (2004) was supported by the numerical study (Kuwano-Yoshida and Asuma 2009). Thus, this paper did not provide a new finding about the development of explosive cyclones.
Moreover, the authors did not conduct quantitative analyses examining the contribution of each factor to the measurements of the intensity of cyclones (e.g., central pressure of cyclones, vorticity). Thus, we cannot judge that the relative importance of each factor to the development of the cyclones.
The authors discussed that the warming by the Kuroshio/Kuroshio Extension weaken the baroclinicity around the Kuroshio/Kuroshio Extension region (P.9, Lines 270-273). However, previous studies (e.g., Hotta and Nakamura 2011 JC, Papritz and Spengler 2015 QJRMS) suggested that the Kuroshio/Kuroshio Extension enhanced the baroclinicity. Thus, the claim of the authors is inconsistent with the previous studies. However, the authors did not discuss about this inconsistency.
Minor comments:
P2 L78:Booth et al. did not study cyclones over the East Asia region.
section 2:The authors should explain the method of the tracking of cyclones in section 2.
section3.2.1:The definition of the baroclinicity is not noted in this manuscript.
Reviewer 2 Report
The manuscript writes well and ideas presented in it are fine, however I do see one major point (point 1 below) that should be addressed.
1) The authors used NCEP final operational analysis. For a study like this that uses data over a 15-year period (2000-2015), operational analysis should definitely NOT be used. The model and data assimilation system has changed a lot over the period of time (see https://www.emc.ncep.noaa.gov/emc/pages/numerical_forecast_systems/gfs/documentation.php). In 2000, the model resolution was T170L42, and in 2015, the resolution was T1534L64. Reanalysis should definitely be used - especially the authors are looking at explosive cyclogenesis which is likely to be strongly affected by model resolution which has changed by close to a factor of 10. They also looked at moisture composites which would also depend strongly on model resolution and parameterization. Because of this, some of their results may not be robust. They should use ERA5, or at least ERA-Interim or JRA 55 or CFSR or MERRA2. Just based on this, I would recommend rejection, because in my opinion, they need to redo all their analyses using a modern reanalysis dataset - which is probably not very difficult given they already have the codes to do that but will take some time. I suggest major revision at this point in the hope that the authors can redo the analysis in time.
2) The authors used a definition of Bergeron that is different from common usage. Perhaps they should name it something different, e.g. modified Bergeron or something like that to avoid confusion.
Reviewer 3 Report
Review of Statistical Characteristics and Composite Environmental Conditions of Explosive Cyclones over the Japan Sea and Kuroshio/Kuroshio Extension (Zhang et al. atmosphere-1334854)
This work focuses on explosive cyclones (ECs) over the Japan Sea and Kuroshio/ Kuroshio Extension and the authors examined the regional characteristics of the two-type ECs around Japan and their related environmental conditions. The several results are consistent with Yoshida and Asuma’s works (2004 MWR) and many. Compared with the previous statistical works, the authors should emphasize what is new findings in the present work. It is useful for the readers to compare with case studies of ECs, to support the validity of the statistical characteristics.
(1) Introduction and conclusion:
The authors should fully explain what is a great advance or improvement from previous statistical studies of ECs in the introduction and conclusion.
L70: KuroshioKuroshio -> Kuroshio/Kuroshio
(2) Data and method:
The authors should describe the detection methods of moving track of EC (how to determine start and end of each track) and the formulation of the cyclonic-vorticity (calculated from the composite wind or geostrophic flow? Relative or absolute vorticity?).
(3) Comparatives in Result
For example, in L191, "shorter than 24h" should be "shorter than or equal to 24h", because Fig. 5 shows ECs of lifetimes of 12, 18, and 24h are 89.1 % of all the ECs (i.e., ECs of 24 h is included in 89.1 %). Please check all the comparatives in the manuscript.
(4) Composite maps, Figs. 8-11
Are the maximum-deepening-rate EC data (L111) used for making Figs. 8-11 (please describe in the figure captions)? Are the anomaly amplitudes significantly large than the standard errors or statistical significant levels for the ensemble averages in the composite maps? The authors can show the standard error or deviation for the anomaly maps, to confirm whether the composite maps are typical for the ECs in the vicinity of Japan, or not.
(5) Composite maps of the surface lows
I do not find the location of the composite surface lows, which were needed to understand the dynamical structure of the composite ECs.
(6) Figure 9, L295-6
In the case study of ECs (Yamamoto 2018 DAO), the humid area is along the cold front over Japan (similar to a1 of Fig 9) and the moist area is extended to warm over the Pacific (similar to a2 of Fig 9). Before the moist flow south of Japan reaches the Japan Sea, the cyclones and cold fronts over the Kuroshio consume the moisture south coast of Japan. This is also one of the possible mechanisms of preventing moisture air over Japan, in addition to the effect of the elevations of the Japanese islands.
(7) L401-3 (interaction between the upper-level and low-level anomalies)
I would like to see the pressure-longitude distributions of PV or vorticity (e.g., Heo et al. 2012 DAO), to confirm the interaction between the upper-level and low-level anomalies.
